# COVID-19-Related Anxiety Symptoms among Quarantined Adolescents and Its Impact on Sleep Pattern Changes and Somatic Symptoms

**DOI:** 10.3390/children9050735

**Published:** 2022-05-17

**Authors:** Yulia Gendler, Ayala Blau

**Affiliations:** The Department of Nursing, School of Health Sciences, Ariel University, Ariel 40700, Israel; ayalabl@ariel.ac.il

**Keywords:** COVID-19, anxiety, sleep disturbance, somatic symptoms, home-quarantine, children, pediatric, adolescents, parents’ perceptions

## Abstract

Background: Home-quarantine due to COVID-19 may have negative psychological effects on vulnerable sub-groups such as children and adolescents. We aimed to explore the prevalence of anxiety among adolescents who were in home-quarantine and its impact on onset of sleep disturbance and somatic symptoms, and on the level of agreement between adolescent and parent perceptions. Methods: Five hundred adolescents (ages 10–17) and 500 parents participated in the study. Adolescents filled out PROMIS Anxiety, PROMIS Sleep Disturbance, and PHQ-15 Physical-Symptom forms, while their parents completed proxy questionnaires containing the same domains. Results: 38% of the adolescents reported experiencing anxiety during home-quarantine period, 29% suffered from sleep disturbance, and 48% reported somatic symptoms. Addition of one day in home-quarantine was significantly associated with sleep disturbance (OR = 3.78, 95%CI: 1.09–8.45) and somatic symptoms (OR = 1.80, 95%CI: 1.01–3.08); female gender was associated with increased risk for somatic symptoms (OR = 2.15, 95%CI: 1.07–4.55); poor agreement in levels of anxiety, sleep disturbance and somatic symptoms was found between adolescent and parent reports (ICCs of 0.197–0.262). Discussion: Total isolation from household members during home-quarantine may cause anxiety, sleep disturbance, and somatization among adolescents. Achieving the appropriate balance between infection control and mitigation of the potential adverse psychological effect of home-quarantine among children and adolescents should be immediate priorities for policymakers.

## 1. Introduction

Acknowledged as a global public health emergency by the World Health Organization (WHO) [1], the first case of COVID-19 in Israel was diagnosed on 26 February 2020 [2]. Health authorities in Israel struggled to contain the spread of COVID-19 by implementing a variety of measures including social distancing, use of masks in public areas [3], and an extensive national vaccination program [4,5]. Another effective measure was self-quarantine [6], which applied to people who may have been exposed to COVID-19 [7]. During the first three waves, these individuals were instructed to subject themselves to household quarantine for a minimum of 14 days, which prevented them from closely interacting with other people from the same household as well as their workplace, schools, and other public venues. The quarantine period was shortened to 10 days for subjects who obtained two negative PCR for COVID-19 test results [6]. 

Several studies have reported a high prevalence of symptoms of psychological distress–including depression and anxiety–among children and adolescents during the COVID-19 outbreak [8,9,10]. A possible explanation for this phenomenon is a combination of biological and environmental factors: A state of stress caused by the COVID-19 pandemic may lead to a disruption in monoamine synthesis in the brain, leading to increased anxiety behavior [11]. Environmental factors such as adverse life events make a substantial contribution to the development of anxiety [12]. Loss of routine, a reduction in social and physical contact, frustration, boredom, and a sense of loneliness caused by COVID-19-based home quarantine may result in increased rates of anxiety symptoms among children and adolescents [9]. 

Anxiety has a complex and bidirectional relationship with sleep disorders [9]. Sufficient sleep is particularly important during childhood and adolescence, when vital physical, cognitive, emotional, and social changes occur. Insufficient sleep can impact daytime function and cause irritability, mood swings, and lethargy [13]. Bates et al. (2020) reported that during the COVID-19 pandemic, children were sleeping more hours during a 24-h period, or alternatively had trouble falling or staying asleep. During home confinement, sleep timing markedly changed (i.e., no set bedtime or wake up time) and lower sleep quality was reported. The increase in sleep disturbance was more pronounced in people with higher levels of anxiety [8,14].

With the increasing severity of the COVID-19 pandemic, somatic symptoms are frequently reported and mostly related to prolonged stress and anxiety [11]. In the general population, anxiety disorders often overlap with somatic symptoms—patients suffering from anxiety express their mental state through somatic symptoms [15]. Studies that focused on mental health of medical workers during the COVID-19 pandemic confirmed these findings and reported that high symptoms of anxiety and severe insomnia are important stressors that may add to somatic symptoms [11]. However, the correlation between anxiety, sleep patterns, and somatic symptoms is still inconsistent. 

The Patient-Reported Outcomes Measurement Information System (PROMIS) pediatric measures were designed to offer feasible and rigorous child-reported evaluation of their physical, mental, and social health, including anxiety and sleep patterns [16,17]. Although self-report of children’s subjective experience is considered paramount, caregivers play a critical role in children’s coping and functioning during the quarantine period [14]. Information provided by proxy respondents is not always equivalent to that reported by the child. Even so, parents’ perceptions of their child’s symptoms should be taken into account, because they may influence healthcare utilization and quality of care provided to the child [18,19]. 

Our study aimed to explore the prevalence of anxiety among adolescents in COVID-19-based home quarantine and its impact on onset of sleep disturbance and somatic symptoms. Secondly, we aimed to examine the level of agreement between levels of anxiety, sleep disturbance, and somatic symptoms reported by adolescents and their parents.

## 2. Materials and Methods

### 2.1. Participants

Five hundred adolescents (age 10–17) who were quarantined due to exposure to COVID-19 positive patients and 500 parents (mothers and fathers, one parent for each child) participated in the study. We chose to focus on this specific age group, because children at these ages can be quarantined in their room without close parental supervision and were able to complete the questionnaire independently without the help of an adult.

### 2.2. Procedure

We conducted a cross-sectional, web-based study during the third wave of COVID-19 in Israel [20], from 20 January to 20 March 2021. We designed two types of self-report questionnaires—one for the adolescents and one for the parent. Adolescents reported their feelings and symptoms according to the domains of the questionnaire, while their parents filled out a proxy questionnaire with the same domains, with reference to their child. The questionnaire took 10–15 min to complete. The URL link for the questionnaires was distributed through Facebook. 

### 2.3. Ethical Considerations

Approval was obtained from the local ethics committee. Written informed consent was received online before the respondents began the questionnaire. Parents were additionally asked to confirm the statement “I permit my child to participate in the study”. Our URL link for the parents contained information regarding the sources of professional advice they may seek for them and their children, including primary physicians, community mental health providers, and urgent care via inpatient and outpatient settings in the hospitals.

### 2.4. Measures

First, demographic characteristics (age, gender, residence) and information regarding the duration of the quarantine were obtained. 

Anxiety level was measured using the 13-item PROMIS Anxiety Short Form that assesses the pure domain of anxiety in children and adolescents [16]. We asked the adolescents to rate the severity of their anxiety during the entire quarantine period. Parents reported their perception of the anxiety of their children using a proxy from. Each item on the measure was rated on a 5-point Likert scale that ranged from 1 (*never*) to 5 (*almost always*). To grade the exact level of anxiety, the total raw scores were converted into T-scores. T-scores of 55 or less indicated no to slight anxiety, 55–59.9—mild anxiety, 60–69.9—moderate, and t-scores of 70 and over indicated severe anxiety. In the current sample, the Cronbach’s alphas of the Anxiety Short Form for children and the proxy form for the parents were 0.95 and 0.90, respectively. 

Self-reported perception of sleep quality, depth, and restoration was measured using the 8-item PROMIS Sleep Disturbance Short Form [13,21]. The form used a 5-point Likert scale that ranged from 1 (*never*) to 5 (*always*), a higher score indicating greater severity of sleep disturbance. To obtain the exact levels of sleep disturbance, the raw scores were transformed into T-scores that were normed on a sample that matched the U.S. general population with respect to age, sex, race/ethnicity, and education [22]. T-scores less than 55 were considered within normal limits, scores of 55–59—mild sleep disturbance, 60–65—moderate, and T-scores of 66 and over indicated severe sleep disturbance [23]. Adolescents ranked their sleep quality within each item, while their parents reported their perception of the sleep quality of their children (using a proxy form). In the current sample, the Cronbach’s alphas of the Sleep Disturbance Short Form for children and the proxy form for the parents were 0.93 and 0.89, respectively.

Prevalence of somatic symptoms was measured using the Patient Health Questionnaire—Physical Symptoms (PHQ-15) [24]. This self-administered questionnaire for children and adolescents was adapted from the PHQ-15 and did not include questions 4 and 11 that are specific to adults [25]. Each item was rated on a 3-point scale that ranged from 0 (*not bothered at all*), 1 (*bothered a little*), and 2 (*bothered a lot*), asking the adolescent to rate the severity of his or her somatic symptoms during the quarantine period. The parents reported their perception of the somatic symptoms of their children using a proxy form. To adjust the level of somatic symptom severity, the total raw scores were prorated to a score out of 30 by multiplying the total raw score by 15 and dividing the value obtained by 13. A score of 0–4 indicated minimal somatic symptoms, a score of 5–9 low level of somatic symptoms, a score of 10–14—medium, and a score of 15–30 indicated high level of somatic symptoms. In the current sample, the Cronbach’s alphas of the PHQ-15 for children and the proxy form for the parents were 0.84 and 0.81, respectively.

### 2.5. Data Analysis

Descriptive statistics were performed to describe the demographic and quarantine-related characteristics of the sample. Chi^2^ test was used to compare between levels of anxiety, sleep disorders, and somatic symptoms as perceived by adolescents and their parents. The associations between anxiety, sleep disturbance, and somatic symptoms were analyzed using Pearson’s correlation. Binary logistic regression was used to analyze the influence of gender, age, population group, area of residence, home-quarantine duration, and anxiety on sleep disturbance and somatic symptoms. Agreement between child and parent scoring was assessed in 2 ways: (1) a comparison of the mean T-scores and (2) intraclass correlations (ICCs). Higher scores of ICC reflect greater agreement among child and parent reports. ICCs of <0.5 were interpreted as a poor level of agreement. The analyses were performed with SPSS Statistics 26 (IBM Corporation, Armonk, NY, USA), and the significance was set at *p* < 0.05. 

## 3. Results

A total of 1000 participants (500 adolescents and 500 parents) were included in the study. The mean age of the adolescents was 13.99 (2.04) years, ranging from 10 to 17. Girls accounted for 53% of the sample. Most of the participants (55%) lived in the central district of Israel. The mean period of home-quarantine was 12.58 (1.36) days, ranging from 10 to 14 days. Among the parents who responded to the questionnaires, 56% were mothers. The mean age of the parents was 43.58 (4.27) years, ranging from 34 to 55 years. Demographic and quarantine-related characteristics of the participants are presented in Table 1.

Levels of anxiety, sleep disturbance, and somatic symptoms were determined using adjusted T-scores and presented in Table 2: 38% of adolescents scored above the threshold of the PROMIS anxiety domain, 29% experienced mild, moderate, or severe sleep disturbance; somatic symptoms were present within 48% of adolescents. The most common somatic symptoms reported by the adolescents were sleep disturbance (42%), feeling tired or having no energy (38%), and headaches (29%). The proxy reports of the parents regarding their children’s anxiety, sleep disturbance, and somatic symptoms were discrepant across all three domains. 

Figure 1a,b demonstrate positive and significant correlations between levels of anxiety and sleep disturbance (r = 0.726, *p* < 0.001) as well as between levels of anxiety and somatic symptoms (r = 0.614, *p* < 0.001) among adolescents who experienced home-quarantine.

Other factors (gender, age, population group, area of residence, home quarantine duration) that could be related to sleep disturbance and somatic symptoms were analyzed using binary logistic regression and presented in Table 3. The most prominent findings of the logistic regression model were: Addition of one day in home-quarantine was significantly associated with sleep disturbance (OR = 3.78, 95% CI: 1.09–8.45) and somatic symptoms (OR = 1.80, 95% CI: 1.01–3.08); female gender was associated with increased risk for somatic symptoms (OR = 2.15, 95% CI: 1.07–4.55); older age decreased the risk for somatic symptoms (OR = 0.73, estimate for 1 year, 95% CI: 0.55–0.94); and anxiety was the most significant predictor for both sleep disturbance (OR = 31.66, 95% CI: 7.46–66.34) and somatic symptoms (OR = 8.44, 95% CI: 2.77–25.72).

Agreement between adolescent and parent reports was assessed in two ways: (1) a comparison of the mean T-scores and (2) intraclass correlations (ICCs). The mean T-scores among adolescents were significantly higher compared to their parents within all three domains, as presented in Table 4. A discrepancy in mean T-scores reflects parental misconception of their children’s anxiety levels, sleep disturbance, and somatic symptoms. Low ICCs (0.262, 0.254, and 0.197) confirmed these findings, indicating poor level of agreement between child and parent reports.

## 4. Discussion

A home-quarantine due to exposure to a COVID-19 positive individual is a necessary restriction for reducing the spread of COVID-19 [6,7]. However, due to isolation from other household members and the threat posed by the pandemic, this represents an extraordinary situation for children and adolescents. Confined during a period of quarantine, this population may experience higher probability of behavioral and emotional disorders, including anxiety [26]. Previous studies showed a prevalence of anxiety symptoms as well as depressive symptoms and posttraumatic stress disorder among children who have been quarantined during pandemic disease [26,27]. Our results indicate that children and adolescents may develop anxiety symptoms even in a short period of home-quarantine.

In our study, sleep disturbance was found among nearly 30% of the adolescents. Complaints regarding sleep disorders were present in both PROMIS ‘sleep disturbance’ and PHQ-15 domains. Sleep disturbance during home-quarantine may be caused by changes in routine, decreased physical activity, absence of social contact with peers, and increase in technology consumption such as mobile phones and computers [8,9]. 

Almost half of the participants reported that they experienced somatic symptoms during home-quarantine. A study by Liu et al. (2020) that was conducted among college and primary school students revealed that stress response to a pandemic as well as decrease in social support may predetermine onset of psychosomatic and somatic symptoms. Similarly, somatic symptoms may appear during the home-quarantine period, which causes prolonged stress among adolescents and prevents continuous support provided by their parents and siblings.

Our present findings are consistent with previous studies demonstrating significant links between anxiety and sleep disturbance [8,9,14], and between anxiety and somatic symptoms [28,29]. Sleep disturbance during home-quarantine is a result of several factors. Isolation and sedentary routine cause sizeable increase in screen time consumption, which has likely impacted sleeping behaviors and delayed sleep times [8]. In addition, a state of anxiety brought on by home-quarantine due to exposure to COVID-19 confirmed cases may significantly contribute to sleep disturbance. Previous studies on the general population in China reported changes in sleep quality, sleep disturbance, and insomnia among participants suffering from anxiety symptoms [26,30]. Similar to our results, a study by Shevlin et al. (2020) determined that a state of anxiety during COVID-19 was associated with increased somatic symptoms within the UK general population. Our study indicated that this association exists among adolescents (age 10–17) during home-quarantine periods in Israel. 

A logistic regression model delineated potential risks and protective factors associated with sleep disturbance and somatic symptoms during home-quarantine: Extension of one day in home-quarantine increases the risk for sleep disturbance and somatic symptoms; female gender predicted increased risk for sleep disturbance; and older age, on the other hand, served as a protective factor for both sleep disturbance and somatic symptoms. The National Child Traumatic Stress Network (2021) highlighted that psychological response to COVID-19 may vary with age. From ages 6–12, higher rates of irritability, sleep and appetite disturbances, somatic symptoms, loss of interest in peers, and excessive attachment to parents may occur. Interestingly, contrary to our present results highlighting possible protective effects of older age, Lavigne-Cerván et al. (2021) reported that among adolescents aged 13–18, sleep problems were more severe, resulting in decreased energy and higher rates of apathy and inattention. Use of electronic resources such as mobile phones and computers contribute to the decrease in quality and hours of sleep among adolescents [9]. This discrepancy might be explained by different study settings and reference periods. While previous research examined sleeping patterns during general quarantine or home confinement, our study focused on a short period (10–14) of home-quarantine, in which adolescents were separated from their parents and the rest of the household members due to exposure to a COVID-19 positive individual to prevent community transmission. In this setting, younger children find themselves for the first time separated from their parents, a situation that can increase their anxiety state and sleep deprivation.

Across all domains, adolescents reported greater levels of anxiety, sleep disturbance, and somatic symptoms than was estimated by their parents. This finding is consistent with a prior study that reported poor agreement between child and caregiver in assessment of anxiety, depression, fatigue, mobility, and pain. However, as opposed to our results, the study revealed that parents highly rated child symptomatology as compared with children’s self-reporting [18]. 

A gap between parent perceptions and children’s reports is anticipated: In an unfamiliar situation of home-quarantine, children do not always know how to communicate their fears and anxieties [31]. Parents may feel pressured in light of the new reality in which they have to cope with their children quarantine process and balance personal life, work, and parenting. This may weaken parents’ ability to be supportive caregivers and violate parent-child relationship [32], leading to poor recognition of child distress and anxiety. Our findings have far-reaching implications, as misconception of children’s perceptions and symptoms may prevent parents from seeking professional help for their children and adolescents.

### Study Limitations

This study had some limitations: The study was conducted online, and the response rate could not be calculated, potentially causing selection bias. Due to the cross-sectional study design, the temporal ordering of the variables could not be established, and so causation between variables cannot be fully indicated. We did not collect information regarding medical background of the participants and were not able to indicate a pre-COVID-19 history of anxiety and mood disorder. 

## 5. Implications

Considering our findings, we recommend that urgent steps be taken to mitigate the impact of home-quarantine on children’s wellbeing. These interventions should target the child and parent as two separate factors. Parents should communicate with children and adolescents to address their fears and concerns and alleviate loneliness and stress, as parent–child discussion was found to be a critical measure in reducing anxiety among children and adolescents [31]. Another important role of the parent is to enforce boundaries with screen time as well as establishing bedtimes. Screen time can be replaced by reading stories over the phone or communicating with extended family members and friends. 

Should anxiety, sleep disturbance, and somatic symptoms due to home confinement increase over time, practitioners will need to be focused on providing resources to maintain the energy caregivers have to sustain positive parenting behaviors and provide sensitive interactions for the children and adolescents that convey a sense of safety and comfort to communicate their fears and concerns with their parents. Parents should be provided with developmentally appropriate educational tools for how to promote healthy habits during home-quarantine and alleviate anxiety.

As spending time in quarantine can take a serious mental toll, some steps that may help mitigate the negative mental health effects are suggested: parents may help their children and adolescents to create a daily schedule; include physical activity that can help combat the sense of malaise and boredom; encourage children and adolescents to communicate with their siblings and friends. If parents identify signs of distress in their children, they should be encouraged to seek professional help online, by phone call, text, email, or through a video call. 

## 6. Conclusions

Our study shows that restrictions related to home-quarantine have far-reaching effects on psychological health for children and adolescents. High prevalence of anxiety symptoms among children and adolescents can manifest in increased levels of physical/somatic problems and sleep disturbance. Furthermore, total isolation from the rest of the household leads to significant discrepancy between child and parent perceptions of the child’s anxiety, sleep disturbance, and somatic symptoms. Achieving the appropriate balance between infection control and mitigation of the potential adverse psychological effect of home-quarantine and empowering well-being in vulnerable groups such as children and adolescents are crucial and should be immediate priorities for policy makers.

## Figures and Tables

**Figure 1 children-09-00735-f001:**
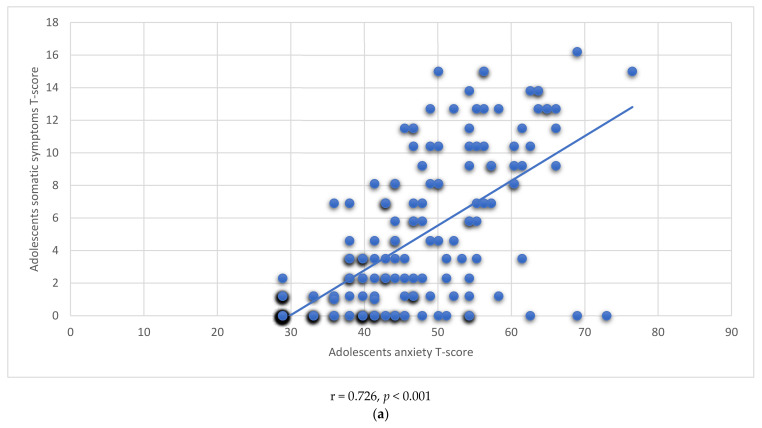
(**a**): Correlation between Anxiety and Sleep Disturbance. (**b**): Correlation between Anxiety and Somatic Symptoms.

**Table 1 children-09-00735-t001:** Demographic and Quarantine-Related Characteristics.

Variable	*n (%) or Mean (SD)*
**Adolescents group (*n =* 500)**	
Gender	
*Male*	235 (47%)
*Female*	265 (53%)
Age (*years*)	13.99 (2.0)
Population group	
*Jews*	410 (82%)
*Muslim-Arabs*	90 (18%)
Area of residence (*districts in Israel*)	
*Jerusalem*	80 (16%)
*Northern*	65 (13%)
*Central*	275 (55%)
*Southern*	80 (16%)
Duration of home-quarantine (*days*)	12.58 (1.4)
**Parents group (*n =* 500)**	
*Mothers*	280 (56%)
*Fathers*	220 (44%)
Age *(years)*	43.58 (4.3)

**Table 2 children-09-00735-t002:** Prevalence of Anxiety, Sleep Disorders, and Somatic Symptoms among Adolescents—Differences between Perceptions of Adolescents and Their Parents—*n* (%).

	Adolescents *n = 500*	Parents *n = 500*	χ^2^	*p*
Anxiety			15.59	0.05
None to slight	310 (62%)	455 (91%)		
Mild	70 (14%)	25 (5%)		
Moderate	90 (18%)	15 (3%)		
Severe	30 (6%)	5 (1%)		
Sleep disturbance			22.88	<0.001
None	355 (71%)	460 (92%)		
Mild	50 (10%)	25 (5%)		
Moderate	65 (13%)	10 (2%)		
Severe	30 (6%)	5 (1%)		
Somatic symptoms			8.85	0.18
Minimal	260 (52%)	385 (77%)		
Low	90 (18%)	70 (14%)		
Medium	125 (25%)	45 (9%)		
Severe	25 (5%)	0		

**Table 3 children-09-00735-t003:** Factors Related to Sleep Disturbance and Somatic Symptoms among Adolescents—a Logistic Regression Model.

	Sleep Disturbance				Somatic Symptoms			
	OR	95% CI		*p*	OR	95% CI		p
		Lower	Upper			Lower	Upper	
Gender								
Male	1 (ref.)				1 (ref.)			
Female	1.07	0.29	3.9	0.91	2.15	1.07	4.55	0.01
Age, estimate for 1 year	0.75	0.56	1.02	0.07	0.73	0.55	0.94	0.02
Population group								
Other	1 (ref.)				1 (ref.)			
Jews	2.74	0.57	12.03	0.21	0.33	0.08	1.32	0.11
Area of residence								
Other	1 (ref.)				1 (ref.)			
Central Israel	2.19	0.68	7.09	0.18	1.66	0.61	4.55	0.31
Home-quarantine duration, estimate for 1 day	3.78	1.09	8.45	0.03	1.8	1.01	3.08	0.01
Anxiety								
No	1 (ref.)				1 (ref.)			
Yes	31.66	7.46	66.34	<0.001	8.44	2.77	25.72	<0.001

OR = Odds Ratio; CI = Confidence Interval.

**Table 4 children-09-00735-t004:** Agreement between Adolescent and Parent Report on Anxiety, Sleep Disturbance, and Somatic Symptom Domains.

	Adolescents Scores		Parents Scores		*t*	*p*	ICC	95% CI	
	Mean T-Score (SD)	Range	Mean T-Score (SD)	Range				Lower	Upper
Anxiety	50.38 (11.69)	33.2–77.0	41.93 (8.78)	32.3–71.3	5.78	<0.001	0.262	−0.028	0.325
Sleep disturbance	47.38 (11.47)	28.9–76.5	41.11 (10.08)	28.9–73.0	4.11	<0.001	0.254	−0.062	0.483
Somatic symptoms	5.49 (4.99)	0–16.2	2.46 (3.46)	0–12.2	4.84	<0.001	0.197	−0.137	0.356

ICC = Intraclass Correlation.

## Data Availability

The data at the basis of the findings of this study are available on request from the corresponding author.

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
