# Peer review of "COVID-19-Related Anxiety Symptoms among Quarantined Adolescents and Its Impact on Sleep Pattern Changes and Somatic Symptoms"

_children, 2022, doi:10.3390/children9050735_

Round 1

Reviewer 1 Report

interesting and well-done study,

please consider if "children" in the title, abstract and main text  is suitable if children aged 10-17 were examined, maybe better "adolescents"?

in the methods section, 2.1 Study group is missing, it is in abstract.

numbering of subsections could be added:

2.2. Measures

2.3. Data analyses

lines 127-128, not Likert scale but ordinal scale, what 1 means?

t-scores or T-scores? they both are used in the manuscript, please unify.

results section and table 1, only one decimal is enough for mean and Sd of age and days:14.0 years, 2.0, 12.6 days, 1.4.

line 166 probably a mistake: "29% experienced mild sleep disturbances", 29% mild or moderate or severe in table 2.

line 167-169 - the presented results are not in any table?

line 196 add "and positive" correlation

line 237 "significantly higher" compared to parents' assessment.

Discussion, lines 254 and 259 on the beginning add "In our study..."

line 323 on the beginning "Should" means "because"?

lines 355-356 and lines 360-362 should be deleted. 

Author Response

Thank you for your time and efforts in reviewing our manuscript. We found the comments and suggestions offered by you to be highly useful.  Accordingly, we incorporated the appropriate changes within the current version of the manuscript. Point-by-point responses to each of the comments are provided below.

Reviewers' comment

Answer

please consider if "children" in the title, abstract and main text is suitable if children aged 10-17 were examined, maybe better "adolescents"?

We modified to 'adolescents' throughout the entire manuscript and the abstract

in the methods section, 2.1 Study group is missing, it is in abstract.

2.1 'Participants' section was added

numbering of subsections could be added: 2.2. Measures 2.3. Data analyses

Numbering of sections was added

lines 127-128, not Likert scale but ordinal scale, what 1 means?

The interpretation of sub-scale 1 was added

t-scores or T-scores? they both are used in the manuscript, please unify.

Was modified to T-scores throughout the entire manuscript

results section and table 1, only one decimal is enough for mean and Sd of age and days:14.0 years, 2.0, 12.6 days, 1.4.

Modified according to your suggestion

line 166 probably a mistake: "29% experienced mild sleep disturbances", 29% mild or moderate or severe in table 2.

Modified according to your suggestion

line 167-169 - the presented results are not in any table?

Lines 167-169 are elaboration of the results in Table 2

line 196 add "and positive" correlation

Modified according to your suggestion

line 237 "significantly higher" compared to parents' assessment.

Modified according to your suggestion

Discussion, lines 254 and 259 on the beginning add "In our study..."

Added according to your suggestion

line 323 on the beginning "Should" means "because"?

Meaning 'in case that…'

lines 355-356 and lines 360-362 should be deleted. 

Thank you for noticing, those lines were deleted

Reviewer 2 Report

Very interesting research. I believe the authors should add the limitations of the dish and the implications to the practice.

Author Response

Thank you for your time and efforts in reviewing our manuscript. We found the comments and suggestions offered by you to be highly useful.  Accordingly, we incorporated the appropriate changes within the current version of the manuscript. Point-by-point responses to each of the comments are provided below.

Reviewers' comment:

Very interesting research. I believe the authors should add the limitations of the dish and the implications to the practice.

Answer:

Limitation chapter is present and was highlighted.

Implication chapter was added.

Round 2

Reviewer 2 Report

I have no comments